# Overexpression of *MsRCI2D* and *MsRCI2E* Enhances Salt Tolerance in Alfalfa (*Medicago sativa* L.) by Stabilizing Antioxidant Activity and Regulating Ion Homeostasis

**DOI:** 10.3390/ijms23179810

**Published:** 2022-08-29

**Authors:** Depeng Zhang, Zhenyue Zhang, Chunxin Li, Yimei Xing, Yaqin Luo, Xinsheng Wang, Donghuan Li, Zhiyun Ma, Hua Cai

**Affiliations:** College of Life Science, Northeast Agricultural University, Harbin 150030, China

**Keywords:** *Medicago saliva* L., *MsRCI2s*, salt tolerance, *H^+^-ATPase*, *SOS1*, *HKT*

## Abstract

Rare cold-inducible 2 (RCI2) genes from alfalfa (*Medicago sativa* L.) are part of a multigene family whose members respond to a variety of abiotic stresses by regulating ion homeostasis and stabilizing membranes. In this study, salt, alkali, and ABA treatments were used to induce *MsRCI2D* and *MsRCI2E* expression in alfalfa, but the response time and the expression intensity of the *MsRCI2D,-E* genes were different under specific treatments. The expression intensity of the *MsRCI2D* gene was the highest in salt- and alkali-stressed leaves, while the *MsRCI2E* gene more rapidly responded to salt and ABA treatment. In addition to differences in gene expression, MsRCI2D and MsRCI2E differ in their subcellular localization. Akin to MtRCI2D from *Medicago truncatula*, MsRCI2D is also localized in the cell membrane, while MsRCI2E is different from MtRCI2E, localized in the cell membrane and the inner membrane. This difference might be related to an extra 20 amino acids in the C-terminal tail of MsRCI2E. We investigated the function of MsRCI2D and MsRCI2E proteins in alfalfa by generating transgenic alfalfa chimeras. Compared with the *MsRCI2E*-overexpressing chimera, under high-salinity stress (200 mmol·L^−1^ NaCl), the *MsRCI2D-*overexpressing chimera exhibited a better phenotype, manifested as a higher chlorophyll content and a lower MDA content. After salt treatment, the enzyme activities of SOD, POD, CAT, and GR in *MsRCI2D-* and *-E*-overexpressing roots were significantly higher than those in the control. In addition, after salt stress, the Na^+^ content in *MsRCI2D-* and *-E-*transformed roots was lower than that in the control; K^+^ was higher than that in the control; and the Na^+^/K^+^ ratio was lower than that in the control. Correspondingly, *H^+^-ATPase*, *SOS1*, and *NHX1* genes were significantly up-regulated, and the *HKT* gene was significantly down-regulated after 6 h of salt treatment. *MsRCI2D* was also found to regulate the expression of the *MsRCI2B* and *MsRCI2E* genes, and the *MsRCI2E* gene could alter the expression of the *MsRCI2A*, *MsRCI2B*, and *MsRCI2D* genes. *MsRCI2D-* and -*E-*overexpressing alfalfa was found to have higher salt tolerance, manifested as improved activity of antioxidant enzymes, reduced content of reactive oxygen species, and sustained Na^+^ and K^+^ ion balance by regulating the expression of the *H^+^-ATPase*, *SOS1*, *NHX**1*, *HKT*, and *MsRCI2* genes.

## 1. Introduction

Alfalfa (*Medicago sativa* L.) is an important perennial legume resource with high nutritional value and yield [1]. However, extreme environmental conditions, especially soil salinization, severely limit its growth and production [2]. Therefore, it is critical to improve its salt tolerance to facilitate its breeding.

High salinity leads to severe ion toxicity in plants. Salt ions, especially Na^+^, once taken up by plant roots are transported to the shoots and eventually accumulate in the leaves, disrupting the cellular ion balance and photosynthesis and impairing enzymatic activity in some metabolic processes [3,4,5]. To cope with sodium toxicity, plants develop complex defense mechanisms, such as up-regulating the expression of membrane ion-transporter genes, which initiates the transportation and compartmentalization of sodium in plant cells [6]. For example, the plasma membrane H^+^-ATPase plays an important role in ion transport and in the establishment and maintenance of ion homeostasis. It rapidly responds to salt stress and is closely related to plant growth and stress tolerance [7]. NHX1 and SOS1 transporter proteins play a role in the translocation of sodium to the vacuole and the efflux of sodium from cells, respectively [8]. Under salt stress, to ensure potassium translocation to alleviate potassium deficiency caused by the high concentration of sodium in plant cell, plants also activate several kinds of potassium transporters, such as HKT [9]. However, the roles of some small membrane proteins that cannot form membrane channels or transporters in response to salt stress are poorly understood.

The rare cold-inducible 2 (RCI2) genes encode a highly conserved small-molecule hydrophobic peptide, identified as a plasma membrane protein by the fused GFP and putative transmembrane domain [10,11,12]. They are similar to Pmp3p in that they regulate ion homeostasis on the basis of similarities in protein structure and location as well as evidence of RCI2 involvement in regulating Na^+^ and K^+^ levels [13,14,15]. Rocha (2016) reviewed the multifunctionality of RCI2s in various plant species and suggested that RCI2s’ function was related to the mechanisms involved in the defense against cold, drought, and salt [16]. Many studies have determined the positive regulatory function of RCI2s in abiotic-stress responses in many species through ectopic expression [17,18,19], mutation [20,21], and functional complementation of RCI2s [22,23]. In addition to regulating ion homeostasis, RCI2s can reduce the treatment-induced accumulation of H_2_O_2_ and malondialdehyde, water loss, and ion leakage and protect cells from oxidative stress due to abiotic stress [24,25].

In alfalfa, Long et al. (2015) identified five RCI2 genes in *Medicago truncatula*: *MtRCI2A*, *MtRCI2B*, *MtRCI2C*, *MtRCI2D*, and *MtRCI2E*. The five RCI2 genes were divided into two groups according to the sequence characteristics of their C-terminal amino acids. *MsRCI2A-C*, homologous to *AtRCI2B*, were classified as Group I, and *M**t**RCI2D-E* homologous to *AtRCI2(D-E)*, containing an extra 20 amino acids in the C-terminal tail, were classified as Group II [26]. MtRCI2(A-C) were found to be localized in the plasma membrane, but MtRCI2D and MtRCI2E fusion proteins were also localized in the intracellular membrane; the difference in the localization results was thought to be related to the extra 20 amino acids in the C-terminal tail of MtRCI2D-E. The *pmp3* yeast mutant complementation experiments were also associated with extra C-terminal hydrophilic tail features. *MtRCI2(A-C)* could functionally complement the salt-sensitivity phenotype resulting from the *PMP3* mutation in yeast, but the *MtRCI2D* and *MtRCI2E* genes failed to complement the *PMP3* deletion in yeast. In previous studies, six RCI2 genes in *Medicago sativa* were identified, and the functions of MsRCI2(A-C) were initially investigated. *MsRCI2(**A-C)* expression was induced under high-salinity, alkali, and ABA treatments, but there were differences between *MsRCI2* gene expression under different treatments. The overexpression of *MsRCI2**(**A-C)* in alfalfa could improve its tolerance to high salinity and mild alkali stress. However, under alkaline stress, *MsRCI2**(**A-C)* exhibited functional differences, which depended not only on their different expression extents but also on the difference in their regulatory relationship with *MsRCI2s* or *H^+^-ATPase* [1].

Owing to the evidence of functional specialization of the RCI2 proteins in different species [11,13], even if *Medicago truncatula* is homologous to *Medicago sativa*, the functional study of MsRCI2s from *M. sativa* is necessary. The aims of this research study were (1) to test the expression and locational differences between MsRCI2E and MsRCI2D under salt, alkali, and ABA treatments, (2) to explore the differences in the response to salt tolerance in *MsRCI2D*- and *MsRCI2E*-transferred alfalfa, and (3) to explain the response mechanism of MsRCI2D and MsRCI2E to salt stress.

## 2. Results

### 2.1. MsRCI2D and MsRCI2E Encode PMP3 Protein and Contain Two Putative Transmembrane Domains

The full-length amplification of *MsRCI2D* and *MsRCI2E* was performed using cDNA as a template to obtain 165 bp and 231 bp fragments. Compared with other *MsRCI2s* and Arabidopsis *AtRCI2* genes, all members of the *RCI2* gene family have conserved the characteristic of PMP3 proteins, containing two putative transmembrane domains (arrows in Appendix A). The amino acid sequence similarity between MsRCI2D and MsRCI2E was 38.89%, with MsRCI2E having an extra 20 hydrophilic amino acids in the C-terminal tail on the second transmembrane domain (TMD). On the basis of the phylogenetic tree (Appendix A), all proteins were divided into four subgroups according to the branching situation: alfalfa *MsRCI2(A-C)* and Arabidopsis *AtRCI2A,-B* belonged to the same branch; *MsRCI2D* and Arabidopsis *AtRCI2C**,-H* belonged to the same branch; and *MsRCI2E,-F* and Arabidopsis *AtRCI2(D-F)* belonged to the same branch. The TMD analysis showed that both proteins contained two TMDs, but the N-terminus of *MsRCI2D* was extracellular, the N- and C-termini of *MsRCI2E* were extracellular; furthermore, *MsRCI2E* contained a hydrophilic C-terminal tail in the second TMD (Appendix A).

### 2.2. Differential Expression of the MsRCI2D,-E Genes under Abiotic Stress

Figure 1 shows the difference in the expression of the *MsRCI2D,-E* genes at different time points under various treatments (salt, alkali, and ABA). In leaves and roots, after salt and alkali stress treatment, the *MsRCI2D,-E* genes were first induced and then decreased, but the expression levels of the *MsRCI2**D,-E* genes were different between the leaves and roots. In addition, under salt stress, the *MsRCI2E* gene was more rapidly up-regulated than the *MsRCI2D* gene. After exogenous ABA treatment, in the leaves, the *MsRCI2D* gene was continuously up-regulated and reached the highest level at 24 h, while in the roots, it reached the highest level at 12 h and gradually decreased. After exogenous ABA treatment, the *MsRCI2D,-E* genes also showed different response patterns in the roots and leaves.

### 2.3. Subcellular Localization of MsRCI2D/-E

As shown in Figure 2, when there was only the GFP, the green fluorescence signal was distributed in the whole cell, but the *MsRCI2D/-E–*GFP fusion protein had signals in the cell membrane. The difference was that the fluorescence signal of *MsRCI2D–*GFP was located only in the membrane, and *MsRCI2E*–GFP had a large number of signals in the whole cell. This showed that the *MsRCI2D–*GFP fusion protein was localized to the cell membrane, while the *MsRCI2E–*GFP fusion protein was distributed throughout the cell, similar to the GFP.

### 2.4. Generation of Transgenic Alfalfa Chimeras That Overexpress MsRCI2D/-E in Hairy Roots via Agrobacterium Rhizogenes

To obtain herbicide-resistant alfalfa chimeras, genetic transformation was carried out with Agrobacterium-mediated methods. The transgenic hairy roots were detected using PCR and real-time PCR (Appendix A). The alfalfa chimeras with a more than 15-fold relative expression level compared with that of the non-transgenic control were considered to be overexpressing and could be used for subsequent research. There were 10 *MsRCI2D-,-E*-overexpressing alfalfa chimeras. The PCR-positive hairy roots were detected using fluorescence microscopy. As shown in Figure 3, no green fluorescence signals were observed in alfalfa hairy roots infected by the empty MSU440, while a green-fluorescence signal was observed in PCR-positive hairy roots.

### 2.5. Overexpression of MsRCI2D/-E Gene Significantly Increases the Salt Tolerance of the Transgenic Alfalfa Chimera

After treatment with 200 mmol·L^−1^ NaCl, the degree of wilting was severe, while the transgenic alfalfa chimera was only partially wilted. With the *MsRCI2E* gene, the degree of wilting in the *MsRCI2D*-overexpressing alfalfa chimera was lower than that in the transgenic alfalfa chimera (Figure 4A). Before treatment, only the chlorophyll content in the *MsRCI2D*-overexpressing alfalfa chimera was significantly higher than that of the WT line (*p* < 0.05). After salt treatment for 5 days, the chlorophyll contents in the alfalfa chimera were 1.45 times and 1.22 times those of the WT line, respectively, i.e., significantly higher than the chlorophyll contents in the WT line (*p* < 0.01 or *p* < 0.05). During the 10-day exposure to salt stress, only the chlorophyll content in the alfalfa chimera transfected with the *MsRCI2D* gene was significantly higher than that in the WT line (*p* < 0.01; Figure 4B). Thus, the overexpression of the *MsRCI2D/-E* gene improved the salt tolerance of alfalfa, and the overexpression of *MsRCI2D* resulted in stronger salt tolerance. Figure 4C shows the changes in relative conductivity under salt stress in the transgenic alfalfa chimera and WT lines. After exposure to salt stress, the relative conductivity of each line increased, while that of the transgenic alfalfa chimera was significantly lower than that of WT lines (*p* < 0.01). Thus, the overexpression of the *MsRCI2D/-E* gene in roots could slow down the increase in leaf cell-membrane permeability and improve the salt tolerance of alfalfa.

### 2.6. Changes in Physiological Indexes in the Transgenic MsRCI2D/-E Gene in Alfalfa under Salt Stress

Figure 4D shows the malondialdehyde (MDA) content in transgenic hairy roots and WT lines under salt stress. After treatment for 5 days, the MDA content in the *MsRCI2E* gene transgenic line was significantly lower than that in the WT line (*p* < 0.01), and there were no significant differences between the *MsRCI2E* gene transgenic line and the WT line. When the samples were exposed to salt stress for 10 days, the MDA content in transgenic hairy roots was significantly lower than that in the WT line (*p* < 0.01). In addition, the soluble-sugar content in transgenic hairy roots was significantly increased and was obviously higher than that in the WT lines (*p* < 0.01; Figure 4E). These results indicate that the overexpression of *MsRCI2D/-E* resulted in less damage to membrane lipids and stronger salt tolerance.

### 2.7. Change in the Antioxidant Capacity of MsRCI2D/-E Transgenic Alfalfa Hairy Roots under Salt Stress

There were no significant differences in the hydrogen peroxide (H_2_O_2_) content in transgenic hairy roots and wild-type lines before treatment and after 5 days of exposure to salt stress. After 10 days of exposure to salt stress, the H_2_O_2_ content in the hairy roots of the *MsRCI2E* transgenic sample was significantly lower than that in the WT line (*p* < 0.01; Figure 5A). In terms of the superoxide anion (oxygen free radical (ORF)) content, when both lines were exposed to salt stress for 5 days and 10 days, the ORF content in each line increased, but the ORF content in transgenic hairy roots was significantly lower than that in the WT line (*p* < 0.05 or *p* < 0.01; Figure 5B), which indicates that when subjected to salt damage, *MsRCI2D/-E-*overexpressing leaves accumulated less H_2_O_2_ and ORF than the WT line and had less oxidative stress damage. Figure 5C–F show the antioxidant enzymatic activities in transgenic hairy roots and WT lines under salt stress. Before treatment, only the superoxide dismutase (SOD) enzyme activity in transgenic hairy roots was significantly higher than that in the WT lines (*p* < 0.05). When both lines were treated with salt stress for 5 days, the enzyme activities of SOD, peroxidase (POD), catalase (CAT), and glutathione reductase (GR) in transgenic hairy roots were significantly increased (*p* < 0.05 or *p* < 0.01) and obviously higher than those in the WT lines. When both lines were exposed to salt stress for 10 days, only the POD activity in *MsRCI2D* transgenic roots and the GR enzyme in *MsRCI2E* transgenic roots were significantly higher than those of the other lines (*p* < 0.01 or *p* < 0.05). The results showed that increased antioxidant enzyme activity played a positive role in reducing the accumulation of reactive oxygen species (ROS).

### 2.8. Changes in Na^+^ and K^+^ Ions in Transgenic Alfalfa Hairy Roots under Salt Stress

To better understand the salt tolerance mechanism in transgenic plants, the changes in the Na^+^ and K^+^ contents and the Na^+^/K^+^ ratio in the roots of the transgenic and WT lines were studied (Figure 6). Before exposure to salt stress, only the K^+^ content in *MsRCI2D* transgenic roots was significantly lower than that in the WT line. After both lines were treated with 200 mmol·L^−1^ NaCl, the Na^+^ content increased and the K^+^ content decreased in each line, resulting in a significant increase in the Na^+^/K^+^ ratio, but the increase in Na^+^ and the decrease in K^+^ in the WT line were greater, leading to a significantly greater increase in the Na^+^/K^+^ ratio in the WT line than in the transgenic line. This indicates that *MsRCI2D/-E* gene overexpression could reduce Na^+^ toxicity under salt stress, increase the K^+^ content in root cells, and ensure the maintenance of the Na^+^/K^+^ ratio, reducing salt damage.

### 2.9. Differences in the Expression of Genes Related to Salt Stress in Transgenic Alfalfa Hairy Roots

To further explore the response of the *MsRCI2D,-E* genes to salt stress, real-time PCR was used to detect the changes in the expression of genes closely related to antioxidants in transgenic hairy roots (Figure 7A–D). After the three lines were exposed to salt stress, the gene expression of *MsCu**/**Zn-SOD* was not significantly different across the three lines. The expression of the *MsCAT* gene in the roots of WT and transgenic lines showed opposite trends. When the three lines were exposed to salt stress for 12 h, the *MsCAT* gene was significantly down-regulated in WT roots, while it was significantly up-regulated in the two transgenic hairy roots, consistent with the change trend shown by CAT enzyme activity. Likewise, the expression trend shown by the *MsGR* gene was different in WT and transgenic lines. Note that under salt stress for 12 h, the gene expression of each line was essentially the same. This result may explain why GR enzyme activity did not change much after the lines were exposed to salt stress. The *GR* gene expression in transgenic alfalfa changed significantly only in the early stages of salt treatment. The expression of the *MsGS* gene was similar to that of the *MsGR* gene, and the gene expression in the roots of each line was the same when the lines were given salt treatment for 12 h. However, it is worth mentioning that under normal conditions, the overexpression of the *MsRCI2E* gene increased the expression of the *MsGS* gene.

Numerous studies have shown that *RCI2* proteins are involved in maintaining cellular ion homeostasis, ATP proton pump (H^+^-ATPase), plasma membrane Na^+^/H^+^ antiporter (SOS1), tonoplast Na^+^/H^+^ antiporter (NHX1), and K^+^ transport channels (HKT1); thus, they also play a role in maintaining cellular ion homeostasis. To determine whether the RCI2 proteins regulate or interact with these membrane proteins, real-time PCR was used to examine the differences in the expression of four genes related to ion homeostasis in transgenic hairy and WT roots (Figure 7E–H). The expression trend of the *H^+^-ATPase* gene was significantly different in the roots of transgenic and WT lines after they were given salt treatment. The overexpression of the *MsRCI2D**,-E* genes significantly increased the *H^+^-ATPase* gene expression; the *H^+^-ATPase* gene was significantly increased (*p* < 0.01) at 6 h of salt stress. However, the *H^+^-ATPase* gene in *MsRCI2D*-transferred roots was continuously up-regulated and that in *MsRCI2E*-transferred roots was obviously decreased. The expression trend of the *SOS1* gene in the two transgenic alfalfa roots was similar, that is, it was significantly increased at 6 h and decreased at 12 h under salt stress, and the up-regulation of the *SOS1* gene was extremely significant in *MsRCI2D* transgenic roots (*p* < 0.01). The change trend shown by the *NHX1* gene in *MsRCI2D* transgenic alfalfa roots was the same as that shown by the *SOS1* gene, while the expression of the *NHX1* gene in WT roots was significantly down-regulated. The expression of the *NHX1* gene in *MsRCI2E* transgenic alfalfa roots did not change significantly but was significantly higher than that in the WT at 6 h (*p* < 0.01). In the roots of WT plants, the expression of the *HKT1* gene first increased and then decreased, but in the roots of the transgenic lines, the expression of the *HKT1* gene decreased. Although changes in gene expression alone are not enough to determine the regulatory relationship between the *MsRCI2D/-E* protein and membrane ion channel proteins, it can be determined that the realization of *MsRCI2D,-E* gene functions under salt stress is closely related to the regulation of ion homeostasis.

### 2.10. Differential Expression of MsRCI2 Genes in MsRCI2D,-E Transgenic Alfalfa

A preliminary analysis showed that both *MsRCI2D* and *MsRCI2E* could respond to salt stress and *MsRCI2(A-C)* could also respond to salt stress, suggesting an interaction between or regulatory relationship within the RCI2 proteins. To test this hypothesis, real-time PCR was used to detect the changes in the expression of the *MsRCI2(A*-*F*) genes in transgenic hairy roots and WT roots (Figure 8). Under normal conditions, the overexpression of the *MsRCI2D* gene decreased the *MsRCI2B* gene (*p* < 0.01), and *MsRCI2E* was up-regulated (*p* < 0.01); the overexpression of the *MsRCI2E* gene decreased the *MsRCI2A* and *MsRCI2B* genes (*p* < 0.05; *p* < 0.01). After salt treatment, the changes in the expression of the *MsRCI2C* and *MsRCI2F* genes in *MsRCI2D,-E* transgenic hairy roots and WT roots were the same (Figure 8C,D), and both showed a downward trend. There were differences between *MsRCI2D/-E* transgenic hairy roots and WT roots in terms of the expression of the *MsRCI2A*, *MsRCI2B*, *MsRCI2D*, and *MsRCI2E* genes (Figure 8A,B,E,F). This indicates that the *MsRCI2D* gene affected the expression of the *MsRCI2B* and *MsRCI2E* genes and the *MsRCI2E* gene affected the expression of the *MsRCI2A*, *MsRCI2B*, and *MsRCI2D* genes.

## 3. Discussion

An increasing number of studies have confirmed the role of RCI2 proteins in abiotic-stress responses of plants genetically engineered to express high levels of RCI2 transgenes (an example of such a response is increased plant resistance to salt, drought, cold, heat shock, osmotic stress, and H_2_O_2_ stress) [17,18]. In alfalfa, the overexpression of *MsRCI2A* transgenic *Arabidopsis thaliana* led to an increased resistance to salt stress [16]. In previous studies, the overexpression of *MsRCI2**(A**-C)* in alfalfa (*Medicago sativ*a L.) not only enhanced salt tolerance but also increased resistance to alkali stress [1]. In this study, *MsRCI2D**/-E* overexpression improved salt tolerance in alfalfa chimeras, manifested as increased chlorophyll and soluble-sugar contents, reduced MDA content, and reduced relative conductivity. However, the notion that *MsRCI2D* and *MsRCI2E* have differences in functions was confirmed by the gene expression under abiotic stress and the phenotypes and physiological indicators of the transgenic plants.

### 3.1. Differences in Terms of Gene Function between MsRCI2D and MsRCI2E under Salt Stress

In [26], six RCI2 proteins from *Medicago sativa* and *Medicago truncatula* were divided into two groups. *MsRCI2A* and *MtRCI2(A-C*) belonged to the first group, both containing about 54 amino acids, and *MtRCI2(D-E*) belonged to the second group, consisting of about 76 amino acids with a C-terminal tail of about 20 amino acids [26]. In this study, MsRCI2D from *M. sativa* had about 54 amino acids, while MsRCI2E had about 76 amino acids, with a hydrophilic C-terminal tail of about 20 amino acids. They were classified into different groups. The subcellular localization of MsRCI2D and MsRCI2E was also different. MsRCI2D–GFP was only located in the cell membrane, while MsRCI2E–GFP had fluorescent signals on the plasma membrane and the inner membrane (Figure 1). This result was not completely consistent with MtRCI2D and MtRCI2E from *M. truncatula.* MtRCI2D–GFP and MtRCI2E–GFP fusions were all located in unspecified intracellular membranes, apparently including the ER [26]. Perhaps this result was related to the 20 C-terminal amino acids of MsRCI2E.

In general, homologous genes in the same species show differences in their expression in response to abiotic stress and hormonal treatment, suggesting that they may play different roles in stress resistance [27,28,29]. In this study, the *MsRCI2D,-E* genes in alfalfa could respond to salt and alkali stress and be regulated by ABA; the expression levels of the *MsRCI2D,-E* genes in roots and leaves showed an overall trend first increasing and then decreasing, but the up-regulation of the *MsRCI2E* gene was more rapid than that of the *MsRCI2D* gene (Figure 2). This indicates that *MsRCI2D,-E* could both respond to salt and alkali stress, but there were differences in the speed of response and the intensity of expression of the two genes.

In this study, *MsRCI2D,-E* overexpression in hairy roots improved the salt tolerance of alfalfa. The analysis showed different trends in antioxidant enzyme activities among different transgenic hairy roots under salt stress. POD and GR activities in *MsRCI2D-* and *-E*-overexpressing hairy roots were significantly different (Figure 6). According to the data results, the antioxidant indexes in hairy roots overexpressing *MsRCI2D* or *MsRCI2E* were not completely consistent; neither was it certain whether this inconsistency was caused by different genes nor could it be determined which gene-overexpressing plants had the strongest antioxidant capacity.

In addition to the differences in gene expression and oxidase activities, there were differences in the regulation of ion homeostasis between MsRCI2D and MsRCI2E. Under normal conditions, only the K^+^ content in *MsRCI2D-*overexpressing transgenic hairy roots was significantly lower than that in the WT line (Figure 7). Correspondingly, the expression of ion channel protein *H^+^-ATPase* and the *SOS1* gene also differed in MsRCI2D- and -E-overexpressing hairy roots (Figure 8). There were also different regulatory relationships between *MsRCI2D,-E* genes, and membrane ion channels.

In conclusion, *MsRCI2D* and *MsRCI2E* have differences in gene structure, response to salt stress, and function, which also reflects the specificity of *RCI2* proteins’ function. The specificity of *RCI2* proteins’ function may be used as a molecular marker of salt tolerance. There are differences in expression or sequence in different alfalfa varieties, and the specificity of this function provides a new idea for the breeding of salt-tolerant alfalfa.

### 3.2. Molecular Mechanism of MsRCI2D and MsRCI2E Regulating Salt Tolerance

At the transcriptional level, the transcription of almost all examined RCI2 genes was altered by one or more stresses, including drought, osmotic stress, low temperature, and salt or upon treatment with ABA, SA, H_2_O_2_, or other signal molecules [16]. In this study, there were differences in the *MsRCI2D,-E* gene expression under salt, alkali, and ABA treatments. Similar studies have been carried out in other species. For example, in *Arabidopsis*, the expression of *AtRCI2* genes was induced by the cold, ABA, dehydration, and high-salinity exposure, and there were differences in their expression [14]. The expression of ten ZmRCI2 members in response to drought stress either induced (*ZmRCI2-1,-2,-6,-7*) or decreased (*ZmRCI2-3,-5,-8,-9,-10*) [30]. These results suggest that the expression of each RCI2 in plant species has an individual characteristic response to its corresponding abiotic stress. RCI2s may not respond to all abiotic stresses but may react to a specific stress type depending on their promoter. Therefore, to understand the functional specificity of RCI2 proteins, further studies need to focus on the upstream-region differences in RCI2s and on the identification of transcription factors that regulate RCI2 expression under specific stresses.

Salt stress can lead to the accumulation of a large amount of ROS in plant cells, and the oxidative stress generated by it destroys various components in the cells, leading to the lipid peroxidation of cell membranes and severe cell damage [3]. Antioxidant enzymes SOD, POD, CAT, GR, etc., are important ROS scavengers that can regulate the active oxygen balance in plants [31]. In this study, the enzymatic activities of SOD, POD, CAT, and GR in transgenic hairy roots with *MsRCI2D,-E* genes (Figure 5) were significantly higher than those of the WT (*p* < 0.05 or *p* < 0.01), indicating that MsRCI2D,-E overexpression could enhance the antioxidant capacity of alfalfa, more effectively remove excess ROS, and reduce membrane lipid peroxidation. This was also demonstrated by the reduced contents of MDA, H_2_O_2_, and OFR. Mohamad et al. found that salt-stress-triggered augmented levels of Na^+^, K^+^, and ROS altered salt-related gene expression in tall wheatgrass [31]. Ion homeostasis is related to ROS scavenging, and plasma membrane transporters play a key role in the response to salt stress [32]. As membrane proteins, RCI2 proteins might alter the changes in Na^+^ and K^+^ and enhance ROS scavenging by regulating other membrane transporters, but the mechanism is still unclear. However, it is certain that *MsRCI2D,-E* overexpression can reduce the ROS accumulation caused by high-salt stress and slow down the damage to plants caused by oxidative stress.

Several studies have shown that RCI2/PMP3 proteins are involved in cellular ion homeostasis, which can stabilize the ion balance in cells and prevent excessive Na^+^ uptake [16,21], as well as regulate membrane stability and polarization. In this study, compared with WT, in transgenic hairy roots, *MsRCI2D,-E* overexpression decreased the Na^+^ content, increased the K^+^ content, and decreased the Na^+^/K^+^ ratio (Figure 6). MsRCI2D,-E played a certain role in cellular ion homeostasis. The expression of *SOS1*, *H^+^-ATPase*, *NHX1*, and *HKT1* genes related to ion transporters or other membrane proteins in transgenic roots also proved this. Due to ion balance stabilization under salt stress, the MDA content in transgenic hairy roots was also significantly lower than that in WT plants (Figure 4D). Although there was no overexpression of exogenous genes in the aerial parts, the electrical conductivity in the leaves was also lower than that in WT plants (Figure 4C). This indicates that the ion balance under salt stress is beneficial to maintaining the stability of the membrane and improving the salt tolerance of plants. Long et al. found that MtRCI2D,-E with an extra 20 hydrophobic amino acids at the C-terminus could not functionally complement the phenotypes of salt sensitivity, resulting from the mutation of *PMP3* in yeasts, and could not regulate salt ionic homeostasis under stress. The function of RCI2 was thought to be directly related to the C-terminal hydrophilic tail. Although MsRCI2E also has an extra 20 hydrophobic amino acids at the C-terminus, the gene still plays a role in regulating ion homeostasis; it seems to be different from MsRCI2D. As mentioned earlier, only the K^+^ content in *MsRCI2D*-overexpressing transgenic hairy roots was significantly lower than that in the WT line. The relationship between the regulation of ionic homeostasis and the 20 C-terminal hydrophobic amino acids is still uncertain and can be demonstrated via mutation in later studies.

RCI2-like genes can improve the salt tolerance of plants by regulating intracellular ion homeostasis, while RCI2 proteins are small-molecule polypeptides and cannot form ion channels alone. Therefore, many studies have speculated that RCI2-like genes may regulate other ion transporters or membrane proteins to function [19,33]. Several studies have shown that H^+^-ATPase can maintain the intracellular pH level under alkaline stress [32,34,35]. PM H^+^-ATPase is a key regulator of NaCl tolerance, as it provides a proton-driving force for Na^+^/H^+^ exchange [36]. Liu et al. found that under salt stress, the activity of H^+^-ATPase in an *Arabidopsis rci2a* mutant, complemented by *MpRCI2* overexpression, was increased compared with its activity in the WT, suggesting that *AtRCI2A* and *MpRCI* genes can affect H^+^-ATPase activity, but the function of RCI2-induced PM H^+^-ATPase activation was not clear [22]. Previous studies have also found a regulatory relationship between *H^+^-ATPase* and *MsRCI2C* under alkali stress, and the expression level of H^+^-ATPase is related to *MsRCI2C*, which maintains intracellular pH [1]. In this study, the expression of the *H^+^-ATPase* gene was only increased in *MsRCI2D*-overexpressing hairy roots at 6 h of salt stress (*p* < 0.01), but it was decreased at 12 h of salt exposure (Figure 7E). However, the relationship between MsRCI2s and PM H^+^-ATPase was uncertain and requires additional analyses of different MsRCI2 proteins when PM H^+^-ATPase is subjected to different stress conditions. The interaction or the trafficking status of MsRCI2s and PM H^+^-ATPase should also be considered in further studies. NaCl-induced CsRCI2E and CsRCI2F interact with the aquaporin CsPIP2;1, reducing the accumulation of CsPIP2;1 on the membrane and reducing water transport [13,19], which is enough to prove that the RCI2 proteins interact with membrane proteins to assist in cellular trafficking.

Most researchers have reasoned that due to their small size, RCI2s cannot act as ion transporters alone, and it is theoretically possible that two or more RCI2 oligomers jointly form a transporter [16]. In a previous study, we found that *MsRCI2A*,B,C gene expression was mutually regulated but physical interactions among MsRCI2 proteins were not concluded [1]. In this study, we also analyzed the expression of the homologous RCI2 genes in transgenic hairy roots with the *MsRCI2D/-E* gene under salt stress. The overexpression of the *MsRCI2D/-E* gene had no effect on the expression of *MsRCI2*C and *MsRCI2F* genes, while the *MsRCI2D* gene could regulate the expression of the *MsRCI2B* and *MsRCI2E* genes, and the *MsRCI2E* gene could affect the *MsRCI2A*, *MsRCI2B*, and *MsRCI2D* genes. Thus, future work should also concern RCI2 oligomers and their function in ion homeostasis and interaction with other proteins. The identification of protein interaction partners of RCI2s is critical to future studies.

At present, it is clear that the overexpression of the *MsRCI2D* and *MsRCI2E* genes could improve the salt tolerance of alfalfa. Combined with existing research, the salt-stress response pattern diagram (Figure 9) was obtained, that is, the *MsRCI2D,-E* genes mainly respond to salt stress by increasing the accumulation of soluble sugars, regulating ion homeostasis, and improving antioxidant systems. It is mainly manifested as the regulation of the *MsRCI2D/-E* gene and membrane ion channel proteins *SOS1*, *H^+^-ATPase*, *NHX1*, and *HKT1*, as well as the mutual regulation with other *MsRCI2* genes, reducing the content of Na+, increasing the influx of K+, and reducing the Na/K ratio, thus maintaining ionic balance. At the same time, due to the change in Na+, the enzyme activities of SOD, CAT, POD, and GR are increased, which promotes ROS scavenging; decreases the contents of MDA, H_2_O_2_, and ORF; and decreases relative conductivity and enhances the antioxidant capacity.

## 4. Materials and Methods

### 4.1. Cloning and Bioinformatic Analysis of MsRCI2 Genes

We designed full-length amplification primers for *MsRCI2D,-E* on the basis of the known nucleic acid sequences of *MtRCI2D* (Medtr6g033495) and *MtRCI2E* (Medtr4g130660), and the primers are shown in Appendix A. Alfalfa cDNA was used as a template. The amino acid sequences of RCI2s in alfalfa and Arabidopsis were obtained from the Phytozome database (https://phytozome.jgi.doe.gov/, for *Medicago truncatula* accessed on 24 April 2014) with the help of TBtools software (V1.09857) (It was freely available at https://github.com/CJ-Chen/TBtools/releases accessed on 1 August 2022) [37]. Alfalfa RCI2-like proteins were analyzed for their conserved motifs and structures compared with the amino acid sequence of Arabidopsis RCI2 proteins. A neighbor joining (NJ) phylogenetic tree was constructed on the basis of the phylogenetic analysis of RCI2-like proteins in alfalfa, Arabidopsis, maize, rice, and cucumber. The pupation of the transmembrane domain of alfalfa RCI2-like proteins was obtained using an online service (http://www.cbs.dtu.dk/services/TMHMM/ accessed on 1 August 2022).

### 4.2. Analysis of the Expression of MsRCI2 Genes under Different Treatments

The seeds of alfalfa “Longmu 806” were selected, and the seedlings were treated with exogenous ABA (100 μmol·L^−1^) and exposed to salt stress (200 mmol·L^−1^ NaCl) and alkali stress (200 mmol·L^−1^ NaHCO3, pH = 8.5) for 20 days. The sampling time points were 1, 6, 12, and 24 h after the start of stress treatment. A total of 200 mg of seedling leaf and root tissue was harvested at each time point and individually stored at −80 °C for RNA extraction. We performed three biological replicates per time point and tissue [1].

An Ultrapure RNA kit (Kangwei Century Biotechnology Co., Ltd., Beijing, China) was used to extract total RNA after abiotic-stress treatment. RNA extracted from cells was reverse-transcribed into cDNA, and the quality of the extracted cDNA was verified via PCR using *GAPDH* as an internal reference gene. Real-time PCR was used to analyze the difference in the expression of *MsRCI2D,-E* genes under salt- and alkali-stress and ABA treatment. The relative expression levels were calculated using the 2^−ΔΔCt^ method. Appendix A lists the gene primers. Three biological replicates (separate experiments) were performed.

### 4.3. Subcellular Localization

To observe *MsRCI2D* and *MsRCI2E* localization in Arabidopsis protoplasts, the target gene was fused with the GFP and constructed on the pUC19 vector. Transient expression [38] was performed in Arabidopsis protoplasts using Arabidopsis Protoplast Preparation and Transformation Kit (Coolaber Co., Ltd., Beijing, China) at room temperature. After 18 days of cultivation without light, fluorescent signals of GFP-labeled MsRCI2s were detected with a laser-scanning confocal microscope (Leica, Heidelberg, Baden-Württemberg, Germany). FM4−64 (Invitrogen, Carlsbad, CA, USA) was used to locate the plasma membrane, and the signal was observed under excitation at 559 nm.

### 4.4. Generation of MsRCI2-Overexpressing Transgenic Alfalfa Chimera

To construct the *MsRCI2* overexpression vector, the MsRCI2D/-E PCR product was connected to pMDC123 via homologous recombination; then, the pMDC123-MsRCI2D/-E plant expression vector was transformed into *Agrobacterium rhizogenes* MSU440-competent cells by the freeze–thaw method.

After the alfalfa seeds were germinated and cultured for 7–8 days, the cotyledonary without roots was pre-cultured as the receptor, and Agrobacterium was activated for infection and co-culture. The cotyledonary was cultured in the dark for about 3–4 days and then placed in sterile water (after adding 100 mg·L^−1^ amoxicillin and clavulanate potassium). This was followed by selection and rooting. The transformed hairy roots were selected using 1.0 mg·L^−1^ glufosinate ammonium. When the glufosinate-resistant hairy roots grew to a length of 3–4 cm, the seedlings were taken out and moved to the greenhouse under controlled conditions [39].

DNA was extracted from the roots of the wild type (WT) and MsRCI2D- and -E-resistant hairy roots, and the DNA of each line was used for PCR detection with *bar*-gene-specific primers (Appendix A). The target gene and the GFP were fused on the MDC123 vector. Therefore, the roots of the wild-type lines and the hairy roots of transgenic alfalfa were intercepted one by one, and green fluorescence was detected with a fluorescence microscope to further identify the transgene-positive hairy roots.

The real-time PCR method was used to quantitatively detect the target gene expression in the hairy roots of the above-mentioned MsRCI2D,-E resistant lines. Appendix A displays the primers of the target gene and internal-reference gene GAPDH primers. Wild-type (WT) lines were used as controls, and all samples were subjected to three biological replicates and three technical replicates.

### 4.5. Determination of Salt Tolerance of Transgenic Alfalfa Chimera

Under growth conditions as above, transgenic and wild type (WT) alfalfa plants were transplanted into separate pots containing vermiculite and perlite in a 1:1 ratio. The plants were watered with a 1/5 concentration of Hoagland nutrient solution every 2 days for 20 days; after that, alfalfa plants in the same growth state were selected for salt treatment (200 mmol·L^−1^ NaCl). In each pot, 200 mL of the relevant solution (1/5 Hoagland solution or 200 mmol·L^−1^ NaCl) was poured, and samples were taken at 0, 6, and 12 days after treatment for physio-biochemical analysis.

The determination of physiological and biochemical indicators was divided into two parts: (1) The relative chlorophyll content (Chl) was measured using a SPAD chlorophyll meter. Each recorded value included 3 biological replicates and 5 technical replicates. The relative conductivity of a leaf blade was then determined with the vacuum method. The malondialdehyde (MDA) content was measured with a UV–vis spectrophotometer and using a Solarbio kit (Beijing Solarbio Science and Technology Co., Ltd., Beijing, China). The MDA content was calculated as the subtraction of absorbance at 532 and 600 nm. The soluble-sugar content in transgenic plants and WT roots after salt stress was measured with a Solarbio kit (Beijing Solarbio Science Technology Co., Ltd., Beijing, China), and the OD value at 620 nm was determined [1,40]. (2) The activities of antioxidant enzymes in WT and transgenic plants were measured after salt stress. The corresponding kits purchased from Suzhou Keming Biotechnology Co., Ltd. (Suzhou, China), were used to determine hydrogen peroxide (H_2_O_2_) and superoxide anion (OFR) contents, superoxide dismutase (SOD) activity, catalase (CAT) activity, peroxidase (POD) activity, and glutathione reductase (GR) activity.

### 4.6. Analysis of Na^+^ and K^+^ Ion Contents in Transgenic Plants

Wild-type and transgenic alfalfa roots were dried to a stable weight at 80 °C. A certain amount of nitrification solution was added. Then, 0.05 g of the material was ground into dry powder, extracted, and filtered in a water bath at 90 °C; then, it was measured using flame atomic absorption spectroscopy. Next, 0, 2, 5, 10, 20, and 40 mL of the mixed standard solution were pipetted into 50 mL volumetric flasks, and equal amounts of K and Na standards were added. Then, the nitrification solution was diluted with deionized water to obtain mixed standard solutions containing K at 0, 2, 5, 10, 20, and 40 μg mL^−1^ and Na at 0, 5, 12.5, 25, 50, and 100 μg mL^−1^. For each strain and time point, three biological replicates were performed after making a standard curve with the prepared mixed standard solution.

### 4.7. Analysis of the Expression Patterns of Related Genes in Transgenic Alfalfa Hairy Roots after Salt Treatment

The analysis of the difference in the expression of related genes in transgenic hairy roots was divided into three parts: (1) Changes in the expression of genes, including Cu/Zn-SOD, CAT, GS, and GR1 genes, related to antioxidant enzymes were detected. (2) To verify the regulatory relationship between MsRCI2s and membrane ion channel transporters, the differences in the expression of H^+^-ATPase, SOS1, NHX1, and HKT1 genes were determined between transgenic and wild-type roots. (3) To determine the regulation between the MsRCI2 family genes, the differential expression of the MsRCI2A-F genes in transgenic and wild-type roots was determined using qPCR. Total RNA extraction and first-strand cDNA synthesis were performed as described above. The expression levels of each gene after 6 h and 12 h of salt treatment were compared with that at 0 h. The primers used are listed in Appendix A.

### 4.8. Statistical Analysis

Three biological replicates were used in all the experiments. All data were organized using Microsoft Excel, 2010; GraphPad Prism version 9.01 (www.graphpad.com/) was used to plot the data; and the one-way ANOVA was used to analyze the data. (*p* < 0.01 represented extremely significant differences, indicated by **; *p* < 0.05 indicated significant differences, indicated by *.)

## 5. Conclusions

MsRCI2D and MsRCI2E were different in their responses to abiotic stress because of their different subcellular localization and expression after salt, alkali, and ABA treatment. The overexpression of the *MsRCI2D,-E* genes improved the salt tolerance of alfalfa, manifested as improved stability of the membrane and activity of antioxidant enzymes, reduced contents of reactive oxygen species and malondialdehyde, and continued ion homeostasis via the regulation of *H^+^-ATPase*, *SOS1*, *NHX**1*, *HKT*, and *MsRCI2* gene expression.

## Figures and Tables

**Figure 1 ijms-23-09810-f001:**
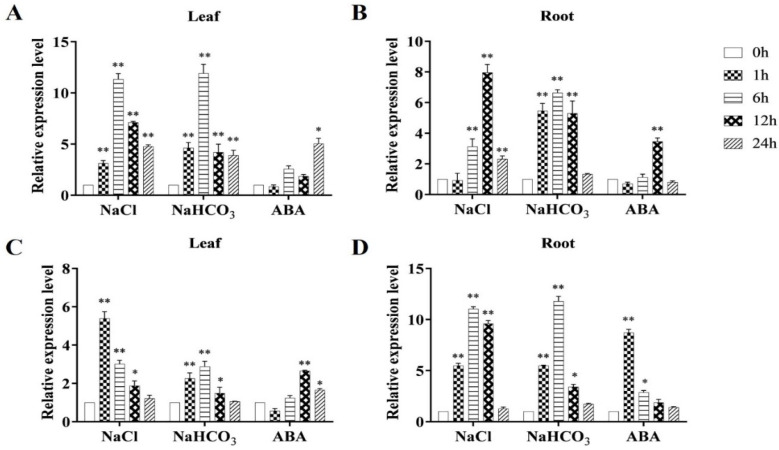
Relative expression of *MsRCI2D* (**A**,**B**) and *MsRCI2E* (**C**,**D**) in leaves and roots under abiotic stress: 200 mmol·L^−1^ NaCl, 200 mmol·L^−1^ NaHCO3 (pH 8.5), and 100 μmol·L^−1^ ABA treatment. The figure on the left shows the gene expression changes in the leaves; the figure on the right shows those in the roots. The values are the means ± SDs of three replicates; * indicates a significant difference (*p* < 0.05); ** indicates an extremely significant difference (*p* < 0.01).

**Figure 2 ijms-23-09810-f002:**
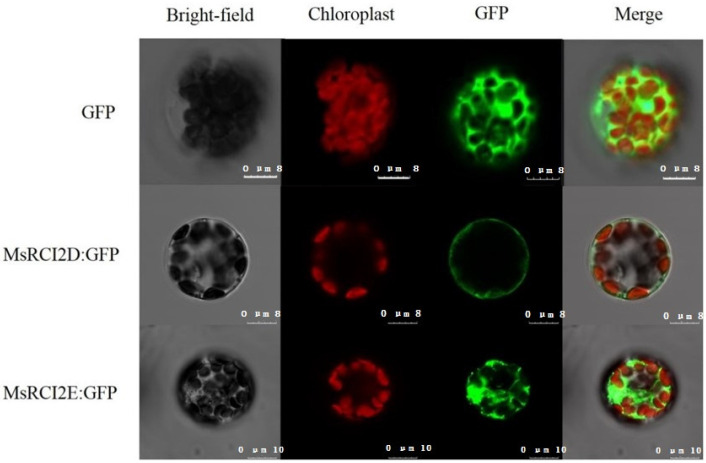
Subcellular localization of MsRCI2D/-E–GFP in Arabidopsis protoplasts. Confocal laser scanning microscope images of Arabidopsis protoplasts expressing GFP fused to MsRCI2D,-E proteins. The red fluorescence signal represents FM4-64-stained chloroplast in a cell. The white scale bars represent 8 or 10 μm.

**Figure 3 ijms-23-09810-f003:**
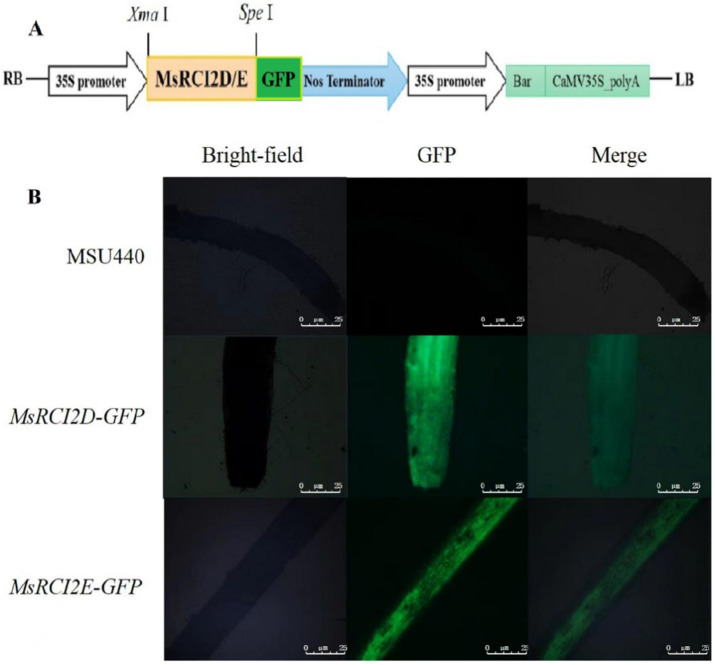
Transgenic alfalfa chimera overexpressing *MsRCI2D–GFP* or *MsRCI2E–GFP* in hairy roots due to Agrobacterium rhizogenes. (**A**) Vector structure; (**B**) green-fluorescent-signal detection in *MsRCI2D/-E* transgenic hairy roots. Confocal laser scanning microscope images of *MsRCI2D-* and *-E–GFP*-overexpressing transgenic alfalfa chimera. White scale bars represent 25 μm.

**Figure 4 ijms-23-09810-f004:**
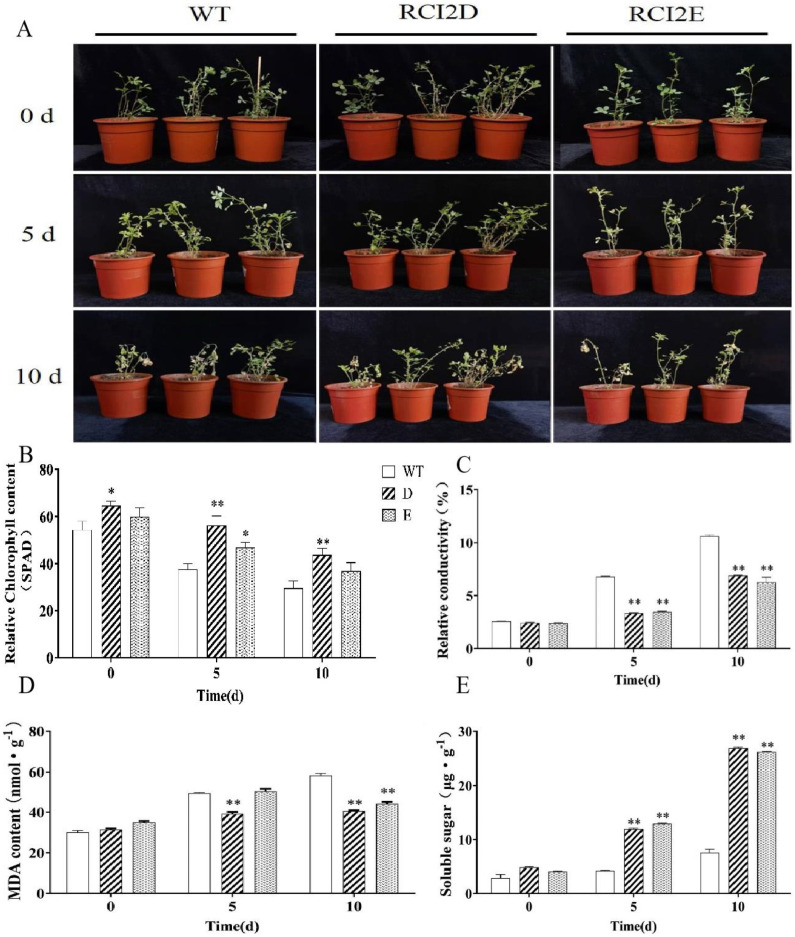
Phenotype (**A**), chlorophyll content (**B**), relative conductivity (**C**), malondialdehyde content (**D**), and soluble sugars (**E**) analysis of alfalfa transformed with the *MsRCI2D* and *MsRCI2E* genes on the 5th and 10th days of salt treatment. WT stands for wild type. RCI2D and RCI2E were transformed with the *MsRCI2D* and *MsRCI2C* genes in hairy roots, respectively. The values are the means ± SDs of three replicates; * indicates a significant difference (*p* < 0.05); ** indicates an extremely significant difference (*p* < 0.01).

**Figure 5 ijms-23-09810-f005:**
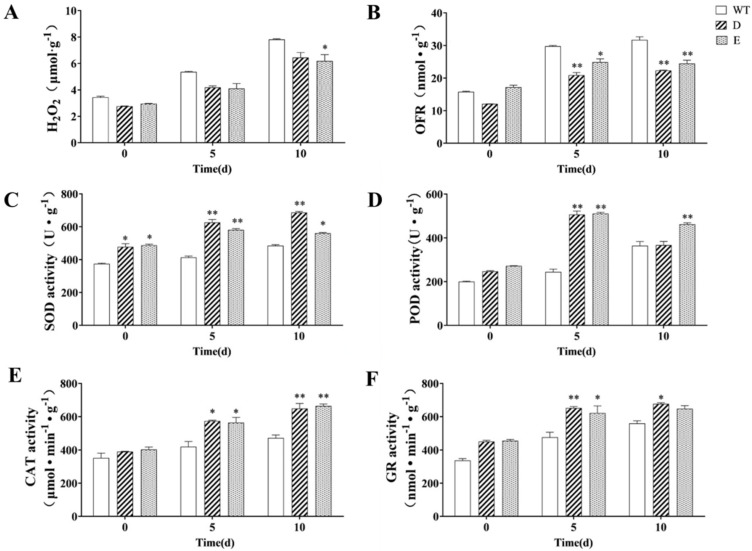
Changes in the antioxidant capacity of MsRCI2D/-E transgenic alfalfa hairy roots before and after exposure to salt stress. Comparison of H_2_O_2_ and ORF contents (**A**,**B**), SOD activity (**C**), POD activity (**D**), CAT activity (**E**), and GR (**F**) in WT plants and transgenic alfalfa hairy roots before and after 5 and 10 days of salt treatment. The values are the means ± SDs of three replicates; * indicates a significant difference (*p* < 0.05); ** indicates an extremely significant difference (*p* < 0.01).

**Figure 6 ijms-23-09810-f006:**
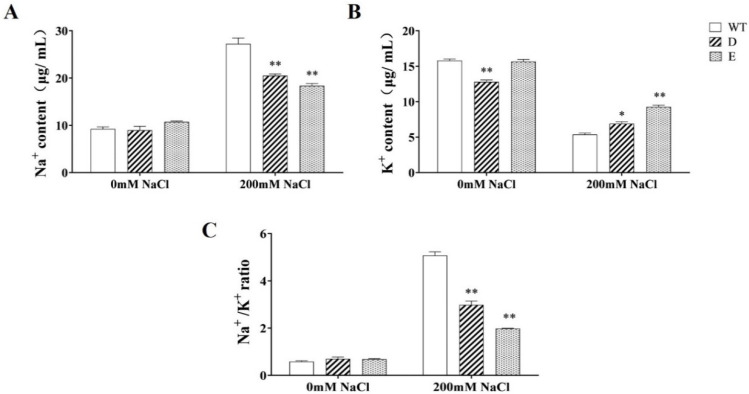
Changes in Na^+^ and K^+^ contents (**A**,**B**) and Na^+^/K^+^ ratio (**C**) in the hairy roots of transgenic alfalfa after salt treatment for 10 days. The values are the means ± SDs of three replicates; * indicates a significant difference (*p* < 0.05); ** indicates an extremely significant difference (*p* < 0.01).

**Figure 7 ijms-23-09810-f007:**
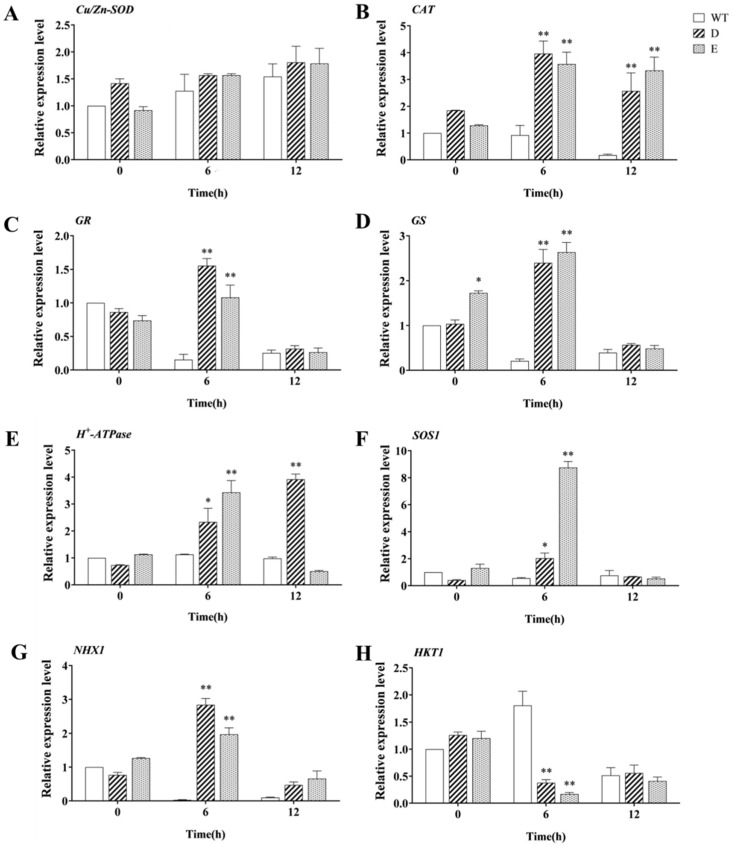
Relative expression of related genes under salt stress in WT and transgenic alfalfa hairy roots. The differences in the expression of genes closely related to antioxidants are shown in (**A**–**D**) ((**A**) *Cu/Zn-SOD*; (**B**) *CAT*; (**C**) *GR*; (**D**) *GS*). The expression of *H^+^-ATPase* (**E**), *SOS1* (**F**), *NHX1* (**G**), and *HKT1* (**H**) was also detected after and before salt stress at 6 h and 12 h. The values are the means ± SDs of three replicates (** *p* < 0.01; * *p* < 0.05).

**Figure 8 ijms-23-09810-f008:**
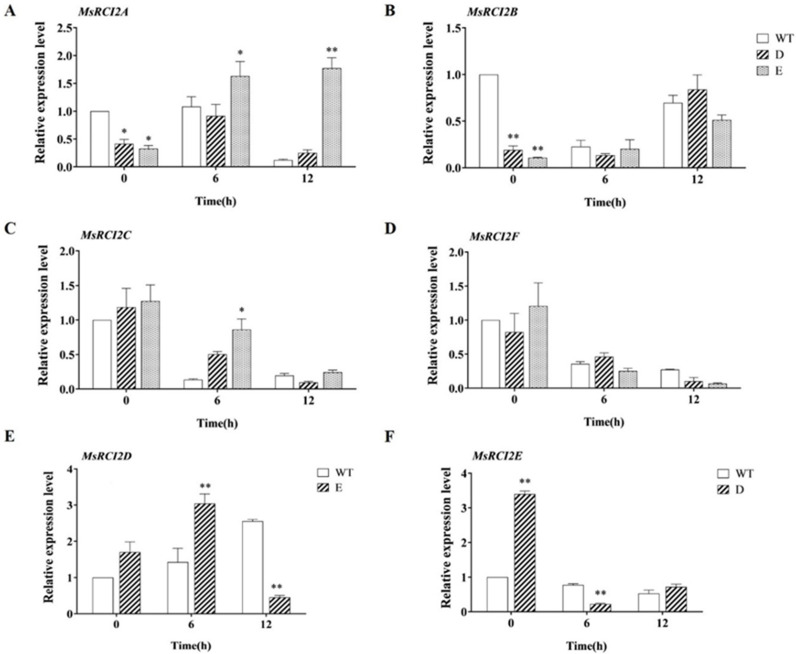
Relative expression of *MsRCI2A-F* genes under salt stress in WT and transgenic hairy roots. (**A**–**D**) Differences in the expression of *MsRCI2A-C* and *MsRCI2F* (**D**) in *MsRCI2D-* and *-E-*overexpressing hairy roots; differences in the expression of *MsRCI2D* (**E**) and *MsRCI2E* (**F**) were also detected. The values are the means ± SDs of three replicates (** *p* < 0.01; * *p* < 0.05).

**Figure 9 ijms-23-09810-f009:**
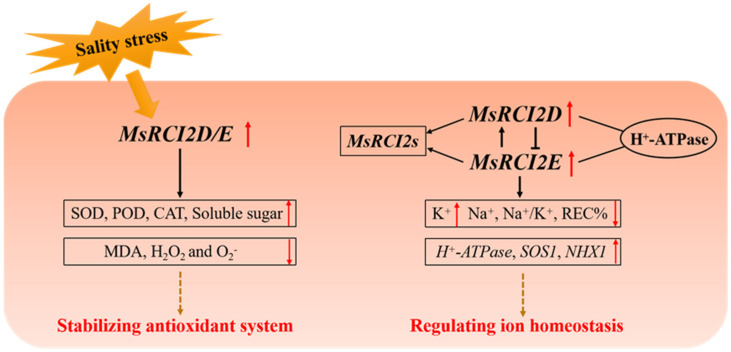
Functional model of the MsRCI2D/-E response to salinity stress. The main functions of MsRCI2D/-E are as follows: (1) ROS scavenging and stabilization of the antioxidant system in response to salinity stress and (2) differences in *MsRCI2* gene expression and regulatory relationships with *H^+^-ATPase*, *SOS1*, *NHX1*, *HKT1*, or *MsRCI2*s.

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
