# Peer review of "Overexpression of *MsRCI2D* and *MsRCI2E* Enhances Salt Tolerance in Alfalfa (*Medicago sativa* L.) by Stabilizing Antioxidant Activity and Regulating Ion Homeostasis"

_ijms, 2022, doi:10.3390/ijms23179810_

Round 1
Reviewer 1 Report
In the present study “Overexpression of MsRCI2D and MsRCI2E Enhances Salt Tolerance in Alfalfa (Medicago sativa L.) by Stabilizing Antioxidant Activity and Regulating Ion Homeostasis”, authors stated that the expression of MsRCI2D, and MsRCI2E in alfalfa was induced by salt, alkali and ABA treatments. I think that the work falls into the scope of the journal and findings are interesting, however MS demands minor revision.
Comments:
Abstract: This section seems scary and unclear. Findings of the study should be clearly defined. Keywords should be different from title.
Introduction: Why study was conducted on this crop? The significant novel point of the study over the precedent studies is not clear. There are two major concerns with this MS. First one is grammatical mistakes, language error, typographical mistakes. Aims and objectives of the study are not clear. There are some related articles that can help the authors i.e. Frontiers in Plant Science 2022, 13: 973782; Emirates Journal of Food and Agriculture, 2021, 33(5): 407-416; Plant Physiology and Biochemistry, 2020, 156: 221-232; and Plant Growth Regulation, 2016, 80: 23-36.
Materials and methods: Please standardize L for liter. What was basis to select these concentrations? Please standardize hr for hours/hour.
Results and Discussion: In results, there is a striking lack of connectors between sentences and leading to confusing. Many sentences are useless in Results section, please focus your key findings. In first use, mention the full names, then you can use abbreviations. Avoid to use “we, our, us”. One way of improving Discussion is to avoid repetition of results in this part. Discussion is very shallow and need in depth discussion with the recent literature published. In discussion, there is a lack of mechanistic approach.
Author Response
Dear Editors and Reviewers:
Thank you for your letter and for the reviewers’ comments concerning our manuscript entitled “Overexpression Of MsRCI2D and MsRCI2E Enhances Salt Tolerance in Alfalfa (Medicago sativa L.) by Stabilizing Anti-oxidant Activity and Regulating Ion Homeostasis”. Those comments are all valuable and very helpful for revising and improving our paper, as well as the important guiding significance to our researches. We have studied comments carefully and have made correction which we hope meet with approval. Revised portion are marked in red in the paper. The main corrections in the paper and the responds to the reviewer’s comments are as flowing:
1.English language changes required.
Response 1: After revising the manuscript,the manuscript had submitted to edit English language by MDPI English Editing Service to revised the language error.
2. Abstract and Keywords
Thank you for your suggestion on the abstract, in which a lot of additions,findings of the study have been defined including changes in antioxidant enzyme activities and gene expression difference of genes related to ion regulation. Keywords have also been modified as “Medicago saliva L.; MsRCI2s ; salt tolerance; H+-ATPase;SOS1;HKT”。
3. Intoduction
(1)Added "Effects of Salt on Alfalfa Production" to explain why alfalfa is studied.
(2)Added "Important role of membrane transporters in salt stress response", explaining the reason for studing on small membrane protein RCI2,and supplemented the suggested 4 useful references.
(3)On the suggestion of “The significant novel point of the study over the precedent studies”,the previous research work was reorganized and revised in the manuscript, highlighting the necessity of the current work.
(4)Referring to the suggested references , in the last paragraph of the introduction, 3 research objectives are clearly listed.
4. Materials and methods
(1)Irregular writing has been revised through language editing services.
(2)On “ select these concentrations”, reference has been incorporated in the corresponding paragraphs in Materials and Methods. As this study is a continuation of previous RCI2A-C study, it is consistent with the approach of the previously published paper [Li,; Song, T.; Zhan, L.; Cong, C.; Xu, H.; Dong, L.; Cai, H.; Overexpression of MsRCI2A, MsRCI2B, and MsRCI2C in Alfalfa (Medicago sativa L.) Provides Different Extents of Enhanced Alkali and Salt Tolerance Due to Functional Specialization of MsRCI2s. Front. Plant Sci.2021,12,702195].
5. result
(1) We have carefully revised the results, adding coherence between sentences, removing redundant and useless descriptions, and elaborating only on important results.
(2)Indeterminate results and inferred statements were removed.
(3)Modified the use of "we our us" sentences. The noun used for the first time, supplemented with full name.
6. Discussion
(1)Repetition of results in this part was deleted.
(2) Referent the recent literature published,Discussion had been revised deeply. Possible reasons for the functional differences between RCI2D and RCI2E are discussed and the focus of future research is listed. The physiological mechanism of RCI2D/E enhancing salt tolerance was discussed from the aspects of ROS scavenging and ion homeostasis regulation. Due to the regulatory relationship between RCI2s genes, it is speculated that it may form ion channels through dimers or participate in the regulation of membrane channel proteins or protein trafficking. Since the research on RCI2 protein in the past 10 years has not been very in-depth, most of the research has only stayed on its role in salt tolerance, and little is known about the molecular mechanism, and the existing results can only infer its possible molecular mechanism, follow-up research will experimentally verify several hypotheses mentioned in the discussion.
Once again, thank you very much for your comments and suggestions.
Reviewer 2 Report
The present manuscript is devoted to the study of the role of RCI2 family genes in responses to salt stress in alfalfa (Medicago sativa L.). The problem of plant resistance to various abiotic stresses, in particular, to salt stress, is extremely relevant, its in-depth consideration and obtaining new experimental data, especially for such important agricultural crops as alfalfa, is of great importance. This work is a logical continuation of a series of experimental studies performed in various laboratories on model plant species (A. thaliana and M. truncatula), as well as a continuation of the authors' work on the analysis of the ectopic expression of RCI2 family genes on transgenic M. sativa plants. This manuscript investigated the role of two genes of this family in a model of chimeric transgenic M. sativa lines in which the formation of hairy roots was induced after treatment with A. rhizogenes explants carrying the gfp gene fused to the target gene (MsRCI2D and MsRCI2E). Using confocal microscopy, different cellular localization of the two proteins encoded by the corresponding genes under study was demonstrated. Their different response to abiotic stress and differences in expression levels after treatment with salts and ABA are shown. The expression of homologous genes in the model of transgenic plants with the induction of hairy roots, which provide overexpression of the MsRCI2D and MsRCI2E genes, was studied, and conclusions were drawn about the need for additional studies related to the identification of their possible regulatory role. An important conclusion of this experimental study is the conclusion about the improvement in the salt tolerance of M. sativa, which was confirmed by some physiological indicators (decrease in reactive oxygen species and malondialdehyde, increase in the activity of antioxidant enzymes, maintenance of ionic homeostasis). The data obtained by the authors make an additional contribution to understanding the mechanisms of realization of plant resistance to abiotic stresses. The submitted manuscript can be recommended for publication in a journal after some clarifications regarding the use of transgenic alfalfa plants with overexpression of the MsRCI2D and MsRCI2E genes.
Remarks: Section 4.5 "Materials and Methods" states that two groups of plants (transgenic and wild-type) were used to determine salt tolerance). According to the results of RT-PCR (Figure 4, Appendix), the lines differed in the expression of the studied genes. The question arises as to which lines made up the experimental group of transgenic plants? It is possible that these lines could be divided into two groups (with high and low expression levels) and compared with each other and with WT plants.

Author Response
Dear reviewer,
Thank you very much for your review and suggestions. Regarding the question of “transgenic alfalfa plants with overexpression of the MsRCI2D and MsRCI2E genes”, make the following responses and explanations.
Genetic transformation was performed by Agrobacterium-mediated methods to obtain herbicide-resistant alfalfa chimeras. Transgenic hairy roots were detected by PCR and real-time PCR (Supplementary Figs 2–4). Because the MsRCI2s gene has endogenous expression, according to the relative expression level of the MsRCI2D/E gene, compared with the non-transgenic control, if the relative expression level exceeds 15, the transgenic alfalfa chimera is considered to be overexpressed and can be used for subsequent studies. In total, there were approximately 20 overexpressing-MsRCI2D/E alfalfa chimeras. In Supplementary Figure 4, not the full result of the transgenic alfalfa roots, but a partial result. The paper we published before is also designed in this way [Li,; Song, T.; Zhan, L.; Cong, C.; Xu, H.; Dong, L.; Cai, H.; Overexpression of MsRCI2A, MsRCI2B, and MsRCI2C in Alfalfa (Medicago sativa L.) Provides Different Extents of Enhanced Alkali and Salt Tolerance Due to Functional Specialization of MsRCI2s. Front. Plant Sci.2021,12,702195]. We did not explain clearly in the manuscript, and the issue of "overexpression" has been supplemented in the revised manuscript.
In addition, in experiments such as physiological indicators and gene expression, the overexpressed alfalfa was randomly mixed with the roots of three seedlings as a biological replicate, which could reduce individual differences between samples. Because we were evaluating the differences between RCI2D ,RCI2E transgenic plants and non-transgenic controls, high and low expression of RCI2D/E were not considered. Perhaps the method you mentioned will be considered in the follow-up in-depth research to compare the difference in function between high and low gene expression level.
Thanks a lot
Hua Cai